# Development of Mouse Hepatitis Virus Chimeric Reporter Viruses Expressing the 3CLpro Proteases of Human Coronaviruses HKU1 and OC43 Reveals Susceptibility to Inactivation by Natural Inhibitors Baicalin and Baicalein

Elise R. Huffman [1,†], Jared X. Franges [2,†], Jayden M. Doster [1], Alexis R. Armstrong [2], Yara S. Batista [1], Cameron M. Harrison [1], Jon D. Brooks [1], Morgan N. Thomas [2], Butler Student Virology Group [1], Sakshi Tomar [3], Christopher C. Stobart [1,4] and Dia C. Beachboard [2,*]

1 Department of Biological Sciences, Butler University, Indianapolis, IN 46208, USA; erhuffman@butler.edu (E.R.H.); ybatista@butler.edu (Y.S.B.); charrison@butler.edu (C.M.H.); jdbrooks@butler.edu (J.D.B.); cstobart@butler.edu (C.C.S.)
2 Department of Biology, DeSales University, Center Valley, PA 18034, USA; jxfrange@ncsu.edu (J.X.F.); aa7849@desales.edu (A.R.A.); mt2467@desales.edu (M.N.T.)
3 National Cancer Institute, Bethesda, MD 20892, USA
4 Interdisciplinary Program in Public Health, Butler University, Indianapolis, IN 46208, USA
* Correspondence: dia.beachboard@desales.edu; Tel.: +1-(610)-282-1100 (ext. 2111)
† These authors contributed equally to this work.

**Abstract:** The recent emergence of SARS-CoV-2 in 2019 has highlighted the necessity of antiviral therapeutics for current and future emerging coronaviruses. Recently, the traditional herbal medicines baicalein, baicalin, and andrographolide have shown inhibition against the main protease of SARS-CoV-2. This provides a promising new direction for COVID-19 therapeutics, but it remains unknown whether these three substances inhibit other human coronaviruses. In this study, we describe the development of novel chimeric mouse hepatitis virus (MHV) reporters that express firefly luciferase (FFL) and the 3CLpro proteases of human coronaviruses HKU1 and OC43. These chimeric viruses were used to determine if the phytochemicals baicalein, baicalin, and andrographolide are inhibitory against human coronavirus strains HKU1 and OC43. Our data show that both baicalein and baicalin exhibit inhibition towards the chimeric MHV strains. However, andrographolide induces cytotoxicity and failed to demonstrate selective toxicity towards the viruses. This study reports the development and use of a safe replicating reporter platform to investigate potential coronavirus 3CLpro inhibitors against common-cold human coronavirus strains HKU1 and OC43.

**Keywords:** coronavirus; reporter virus; common cold; HKU1; OC43; antivirals

## 1. Introduction

Coronaviruses (CoVs) are enveloped positive-strand (+ssRNA) viruses that include several prominent animal and human pathogens [1]. There are currently seven known human coronaviruses that, collectively, are responsible for upper and lower respiratory diseases of varying severities [2]. More recently, several of these coronaviruses have emerged that have demonstrated the potential for emerging coronaviruses to trigger widespread disease associated with both high hospitalizations and mortality rates in infected populations. The SARS-CoV epidemic of 2002–2003 rapidly spread to 29 countries, infecting about 8000 individuals with approximately 10% mortality [3,4]. The emergence of MERS-CoV, a decade later in 2012, was associated with a case fatality rate of nearly 35% [5,6]. Most recently, in December 2019, a SARS-CoV-like human coronavirus emerged in Wuhan, China, and has spawned a worldwide pandemic associated with COVID-19 [7]. While much attention has been drawn to the impacts of coronaviruses on human public health, there

continue to be outbreaks of animal coronavirus infections, including the porcine epidemic diarrhea virus (PEDV) [8]. This virus has a very high mortality rate of 80 to 100% in piglets and has a great economic impact on the pork industry [8,9]. Collectively, these recent events demonstrate the high potential for known and emerging coronaviruses to cause significant disease in humans and domesticated animals and the importance of continued research into both vaccine and therapeutic options.

While we have recently successfully developed vaccines to SARS-CoV-2 in response to the pandemic, there are no vaccines for other coronaviruses, and there remains an ongoing need to continue the development of effective antivirals to treat existing and future coronavirus infections [2,10]. The development of coronavirus antivirals continues to pose several distinct challenges [11,12]. First, the increasing recognition of zoonotic coronaviruses creates new targets that may need to be individually addressed [13,14]. Second, many coronaviruses are difficult to cultivate *in vitro*, and several of them lack robust reverse genetic or biochemical systems for the testing of antivirals [6]. Third, the cultivation and replication conditions of different coronaviruses *in vitro* vary dramatically, making direct comparisons difficult. Finally, human, animal, and potential zoonotic coronaviruses now occupy a large number of divergent genetic lineages, between which there is a substantial difference in even primary amino acid sequences in viral proteins and enzymes [15]. Thus, the challenge of identifying broadly active inhibitors and defining their mechanisms is significant. Most efforts to develop coronavirus antiviral therapeutics to date have targeted the viral replication complex machinery (including the viral RdRp) and the virus-encoded cysteine proteinase activity in the coronavirus nonstructural protein 5 (nsp5), also known as the 3C-like protease (3CLpro) or main protease (Mpro) [11]. All coronaviruses express the 3CLpro protease as one of up to sixteen nonstructural proteins (nsps) included in the large replicase polyproteins that are translated from the input positive-strand RNA genome (+ssRNA) (Figure 1) [1,16]. Upon translation, 3CLpro is required for the processing of up to 11 cleavage sites between nsps 4 and 16 during assembly of the viral replication complex.

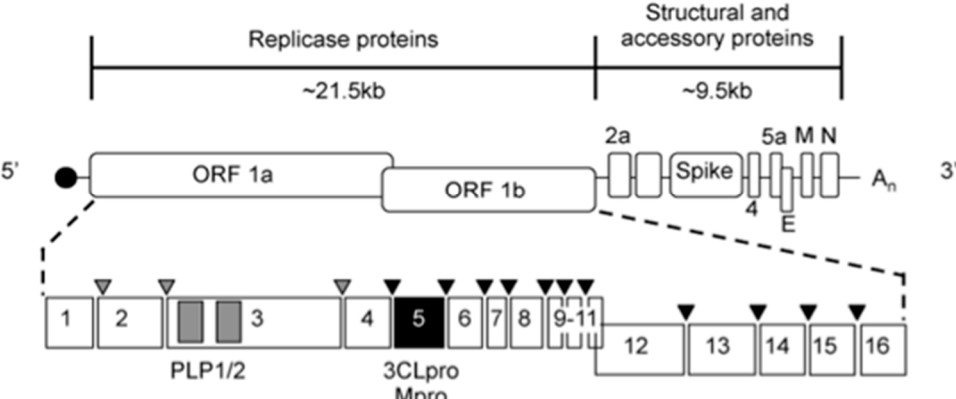

**Figure 1.** MHV genome, replicase polyprotein expression, and 3CLpro-mediated processing. The approximately 31 kb MHV genome is shown at the top with the 5′ cap and 3′polyadenylated tail ($A_n$) and seven genes. The open reading frame 1 (ORF1a/b) is translated into a large polyprotein that contains 16 nonstructural proteins (nsp1-16). The nsps are proteolytically processed by papain-like protease domains (PLP1/2) within nsp3 (grey) and the 3CLpro (or Mpro) encoded as nsp5 (black). E, envelope; M, matrix; and N, nucleocapsid.

The crystal structures of 3CLpro proteases have been solved for several human coronaviruses (including SARS-CoV, SARS-CoV-2, and HCoV-HKU1) that have aided researchers in the targeted design of potential antivirals [17–19]. Coronavirus 3CLpro consists of three domains with domains 1 and 2 forming a chymotrypsin-like fold containing the Cys-His catalytic dyad and active site and the mostly helical domain 3 playing roles in both 3CLpro dimerization and regulating 3CLpro activity [20,21]. Several SARS-CoV-2 3CLpro inhibitors have been identified from plants used in traditional Chinese medicinal prepara-

tions and teas [22]. Baicalein and baicalin are two of the most abundant compounds in the Shuanghuanglian preparation that is commonly used to treat respiratory infections [23]. Andrographolide is a component found in the *Andrographis paniculata* plant indigenous to South and Southeastern Asia that is often used to treat upper respiratory infections and has been shown to target SARS-CoV-2 3CLpro [24,25]. It is unknown whether these compounds are active against other human coronaviruses.

We previously recovered recombinant MHV chimeras expressing the 3CLpro of the closely related human subgroup 2a betacoronaviruses HCoV-HKU1 (H5-MHV) or HCoV-OC43 (O5-MHV) [16]. HCoVs HKU1 and OC43 are two of four common circulating strains of coronaviruses associated with seasonal common colds. In this study, we developed luciferase reporter strains of H5- and O5-MHV for the screening of novel therapeutics. We demonstrate that chimeric reporter viruses replicate similarly to wild-type (WT) MHV and produce luciferase with similar kinetics to the previously reported MHV-FFL. Next, we tested three natural compounds (baicalein, baicalin, and andrographolide) that have been shown previously to inhibit the SARS-CoV-2 3CLpro for activity against the HKU1 and OC43 3CLpro proteases expressed in the background of MHV. We found differences in the levels of inhibitory activity among both inhibitors and virus strains. This work reports the successful recovery of chimeric reporter viruses that can be used to rapidly screen and identify compounds that are broadly effective against nsp5 proteases.

## 2. Materials and Methods

### 2.1. Viruses, Cells, and Inhibitors

Recombinant MHV A59 (GenBank accession number AY910861) virus was used as a WT control virus. Recombinant strains WT-FFL, O5-MHV (O5), and H5-MHV (H5) were also used and have been previously described [16,26]. Delayed brain tumor cells (DBT-9), a murine astrocytoma cell line, were used for virus cultivation and were grown in Dulbecco's Modified Eagle Medium (DMEM; Gibco, Grand Island, NY, USA) supplemented with 10 mM HEPES (Gibco), 10% heat-inactivated fetal bovine serum (R&D Systems, Minneapolis, MN, USA), and an antibiotic–antimycotic mixture containing penicillin, streptomycin, and amphotericin B (Corning, Glendale, AZ, USA). Additionally, baby-hamster kidney 21 cells expressing the MHV receptor (BHK-R) were used during the initial recovery of the chimeric H5- and O5-FFL viruses described below. These cells were cultured in complete DMEM supplemented with 0.8 mg/mL of G418 (CellGro, Lincoln, NE, USA) to maintain the expression of the murine carcinoembryonic antigen cell adhesion molecule 1 (CEACAM1) receptor. Both DBT-9 and BHK-R cell cultures were incubated at 37 °C under 5.0% $CO_2$. The inhibitors tested were baicalein (TCI America, Portland, OR, USA), baicalin (Synaptent LLC, Chicago, IL, USA), and andrographolide (TCI America), and stocks were prepared and stored using the manufacturer's recommendations until use.

### 2.2. Virus Recovery of H5-MHV-FFL and O5-MHV-FFL

Chimeric MHV viruses were engineered to express both a firefly luciferase (FFL) reporter as an nsp2-fusion and either the nsp5 (3CLpro) of HCoV-HKU1 (named H5-MHV-FFL) or HCoV-OC43 (O5-MHV-FFL). Genetic changes to independently assemble the WT-MHV-FFL-nsp2 fusion (MHV-FFL) and the H5 and O5 chimeric viruses have previously been reported [16,26]. In this study, viruses containing the H5-FFL and O5-FFL were generated using the reverse genetics system for MHV-A59 [27]. Briefly, the MHV-A59 genome was divided into seven cDNA fragments, which were digested using the appropriate restriction enzymes. Specifically, fragments containing both the FFL and heterologous nsp5 (3CLpro) proteases as well as the remaining MHV genomic fragments were ligated at 16 °C overnight before the DNA was purified, *in vitro* transcribed, and electroporated into BHK-R cells along with N gene transcripts. The electroporated cells were co-cultured with the DBT cells and incubated at 37 °C until the induction of a cytopathic effect (CPE). The virus produced from the electroporated cells (passage 0 [P0]) was passaged onto uninfected DBT cells to generate a P1 stock virus that was used for all the experiments.

### 2.3. RT-PCR and Sequencing

The P0 virus was sequenced across FFL and nsp5 to ensure that no additional mutations were present. The total intracellular RNA was isolated using TRIzol (Invitrogen, Waltham, MA, USA), following the manufacturer's protocol. Viral RNA was then reverse-transcribed using Superscript III (Invitrogen) and random hexamers (Roche, Basel, Switzerland). The FFL and nsp5 coding sequences were amplified by PCR using primers (IDT) complementary to nucleotides 596 to 613 (sense; 5′-ATGGAAGACGCCAAAAACATAAAGAA AG-3′) and 1257 to 1273 (antisense; 5′-CGGGGGAGTCTTTTAAC-3′) for FFL and nucleotides 10155 to 10177 (sense; 5′-GTTAAAAGACTCCCCCG-3′) and 11783 to 11799 (antisense; 5′-GGAGTGACAAGATTCCC-3′) for nsp5 (the genomic locations are listed in reference to their position in the WT-MHV genome). The FFL and nsp5 amplicons generated were directly Sanger-sequenced to analyze them for retention of the engineered mutations and the absence of additional mutations. Passage mutants obtained in the presence of baicalein were sequenced using the RT-PCR methods described above after RNA isolation using a viral RNA isolation kit (Zymo Research, Orange, CA, USA).

### 2.4. Viral Replication Assay and Luciferase Assay

DBT cells were infected with either WT, H5, O5, WT-FFL, H5-FFL, or O5-FFL viruses at an MOI of 1 and absorbed for 30 min. The cells were then washed twice with PBS, the media were replaced, and the cells were incubated at 37 °C. The supernatants were sampled over time, and the viral titers were determined by plaque assay, as previously described [21]. The cells were lysed in reporter lysis buffer and analyzed for luciferase activity following the manufacturer's protocol (Promega, Madison, WI, USA) on a luminometer.

### 2.5. Cytotoxicity Analysis

Cytotoxicity was determined using a trypan blue exclusion assay. DBT-9 cells were cultured in 12-well plates and treated with a range of inhibitor concentrations from 0 to 100 μM. After a 24 h incubation period, the cells were harvested and live cell counts determined using trypan blue stain exclusion.

### 2.6. Inhibitor Assays

To test for inhibition of virus replication in the presence of baicalein, baicalin, or andrographolide, a luciferase assay was performed. DBT-9 cells were treated with inhibitors six hours before, at the time of, or six hours after infection with either H5-FFL or O5-FFL at an MOI of 0.01. At 12 h post infection (h p.i.), the cells were harvested through cell scraping and frozen at −80 °C to ensure complete cell lysis. The harvested samples were thawed and plated on a white 96-well plate, luciferin (Promega, Madison, WI, USA: Goldbio, St. Louis, MO, USA) was added, and luciferase activity was measured on a luminometer.

### 2.7. Analysis of Viral Escape

The viruses were passaged in the presence of $3\times$ the experimental $EC_{50}$ of the inhibitor for a total of five passages to analyze viral escape mutations. Confluent flasks were infected with respective viral stocks and media were replaced with inhibitor-containing media. For each passage, DBT-9 cells were infected for 12 h, and the cells were subjected to a freeze–thaw procedure before passaging to the next cells. After the fifth passage, viral RNA was extracted as described above from the supernatants and sequenced across the nsp5-coding region and analyzed for mutations.

## 3. Results

### 3.1. H5-MHV and O5-MHV Chimeric Reporter Viruses Expressing Firefly Luciferase (FFL) Do Not Exhibit Altered Replication Kinetics

To aid in the efficient screening of potential inhibitors of 3CLpro, we sought to establish luciferase reporter versions of each of these chimeric MHV viruses (Figure 2). We previously reported the recovery of recombinant chimeric mouse hepatitis virus (MHV) strains that

encode the 3CLpro proteases of subgroup 2a human β-CoVs, HCoV-HKU1 (H5-MHV), and HCoV-OC43 (O5-MHV) [16]. The incorporation of these heterologous coronavirus proteases into MHV was well tolerated with indistinguishable replication kinetics compared to WT-MHV. To these chimeric viruses expressing either HKU1 or OC43 3CLpro, a firefly luciferase gene sequence was engineered as an in-frame fusion to nonstructural protein 2 (FFL-nsp2), since we had previously shown that FFL-nsp2 fusion was tolerated by MHV and it replicates indistinguishably from the WT virus [26]. The resulting viruses named H5-MHV-FFL and O5-MHV-FFL were successfully recovered and exhibited a plaque morphology indistinguishable from WT-MHV as well as H5-MHV and O5-MHV. Both viruses were recovered and sequenced across the nsp1-FFL-nsp2 junction and across nsp5, revealing only engineered changes, intact FFL, and chimeric nsp5 with no additional mutations.

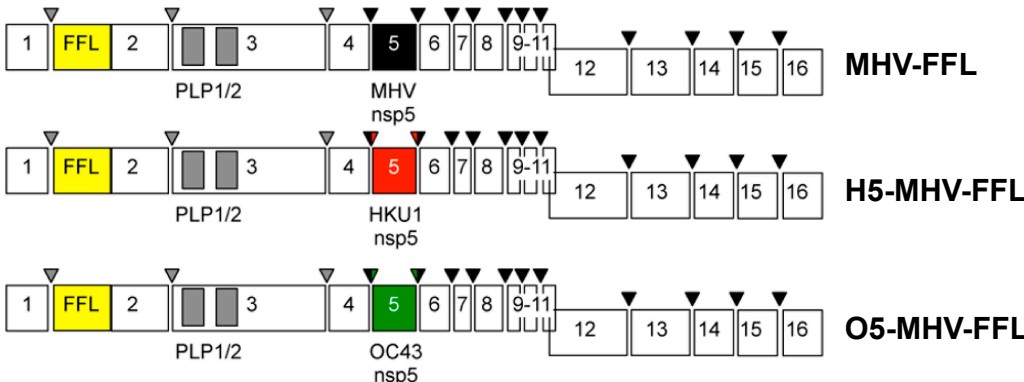

**Figure 2.** Engineered H5-MHV-FFL and O5-MHV-FFL reporter viruses. Schematics of the reporter viruses are shown with firefly luciferase, FFL (yellow), and chimeric nsp5 (3CLpro) proteases from HCoV-HKU1 (red) or HCoV-OC43 (green). PLP, papain-like protease.

To evaluate and compare replication kinetics, DBT-9 cells were infected with WT-MHV, WT-MHV-FFL, H5-MHV, H5-MHV-FFL, O5-MHV, and O5-MHV-FFL at an MOI of 1 for 24 h, and the supernatant virus titer was determined by plaque assay (Figure 3A). There were no significant differences observed between either WT MHV or O5-MHV and their FFL-encoding homologs (MHV-FFL or O5-MHV-FFL) at each time point (Student's *t*-test, $p < 0.005$). However, there were significant differences observed between H5-MHV and H5-MHV-FFL at 0, 4, and 8 h post infection (h p.i) based on the virus titer (Student's *t*-test, $p < 0.005$). Despite these differences, each virus reached peak infection with comparable titers at the same time (12 h p.i.).

We next compared the FFL relative light units (RLUs) measured in the same monolayers used to determine the virus titer (Figure 3B). The FFL-RLU measurements showed no significant differences at each time point between FFL-encoding strains (Student's *t*-test, $p < 0.005$). In addition, each virus reached peak signal at 10–12 h p.i, consistent with the peak infection obtained through titering. WT MHV not encoding FFL was measured at 16 h p.i. to illustrate the background signal of the assay and exhibited a low average of 69 RLUs. Collectively, the titer and luciferase data indicate that the use of FFL-derived RLUs is a robust surrogate for virus replication, both in timing and peak replication.

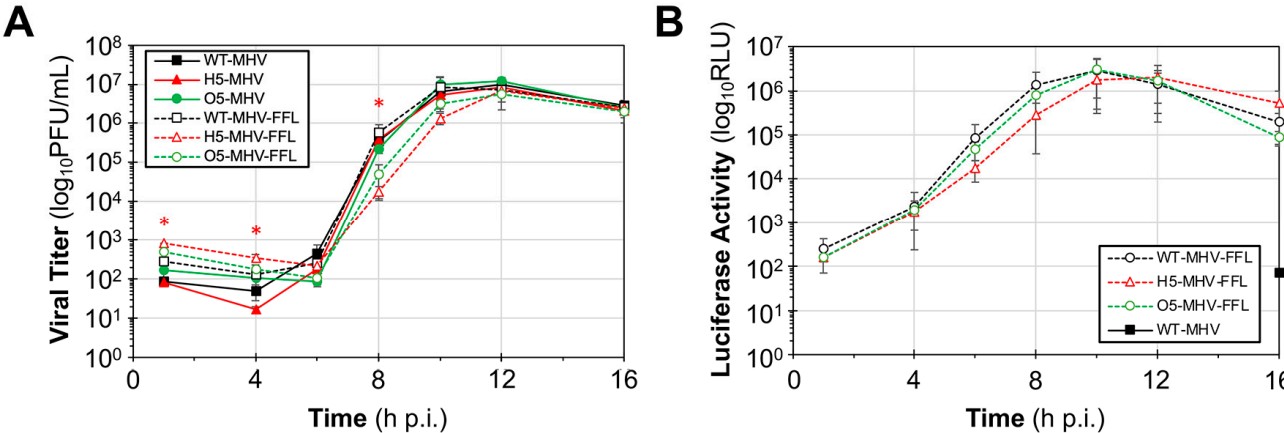

**Figure 3.** Replication and luciferase kinetics of chimeric reporter MHV viruses. (**A**) DBT-9 cells were infected with the indicated viruses at an MOI of 1 PFU/cell; supernatants were sampled over 24 h and titered by plaque assay. The average titers (±s.d.) of three experimental replicates are shown. Statistical analyses (pairwise Student's *t*-test) were performed between conventional (lacking FFL) and reporter viruses (containing FFL) at each time point. Significant differences ($p < 0.005$) are denoted by colored asterisks (*). (**B**) DBT-9 cells were infected with the indicated viruses at an MOI of 1 PFU/cell. At times indicated, cells were lysed in reporter lysis buffer, and luciferase activity was measured in relative light units (RLU). The average luciferase activity (±s.d.) of three experimental replicates is shown. Statistical analyses (Pairwise Student's *t*-test) were performed between each virus at each time point. No significant differences ($p < 0.005$) were observed.

### 3.2. Analysis of Cytotoxicity of Natural Inhibitors of SARS-CoV-2 3CLpro

Three different natural plant products (baicalin, baicalein, and andrographolide) have been previously identified as inhibitors of the 3CLpro protease of SARS-CoV-2 [24,28]. Before testing these compounds for inhibitory activity against H5-MHV-FFL and O5-MHV-FFL, each of these compounds were evaluated for cytotoxicity in DBT-9 cells.

Baicalein and its related glucuronide analogue, baicalin, were each evaluated for cytotoxicity in DBT-9 cells using a trypan blue exclusion assay (Figure 4A,B). Both baicalein and baicalin exhibited very little cytotoxicity at concentrations below 100 μM and similar cytotoxic effects at higher concentrations. At the highest concentration tested (1500 μM), both compounds were associated with significant reductions in cell viability with very few living adherent cells remaining. The concentration associated with 50% reductions in viability ($CC_{50}$) was calculated to be 379.1 μM for baicalin and 224.4 μM for baicalein. However, there were no significant differences observed when comparing the two treatments at each concentration (pairwise Student's *t*-test, $p < 0.005$).

Andrographolide is a diterpenoid that differs considerably in its structure from both baicalin and baicalein (Figure 4C). Andrographolide was also evaluated for cytotoxicity in DBT-9 cells. Reduced cell viability was detected at all concentrations evaluated for andrographolide (Figure 4D). The $CC_{50}$ for andrographolide was calculated to be 70.4 μM. At even the lowest concentrations tested (1.56 μM), cell rounding and reduced cell attachment were visibly observed in DBT-9 cells treated with andrographolide (Figure 4E). At the highest concentration tested (100 μM), the cells had largely undergone apoptosis, and only small cell fragments or remnants remained visibly adherent. Due in large part to the visible morphological changes present at all tested concentrations, the appreciable cytotoxicity of andrographolide, and concerns that impacts on cell viability and morphology could affect viral replication, we decided to continue with assessing baicalin and baicalein for antiviral activity and not andrographolide.

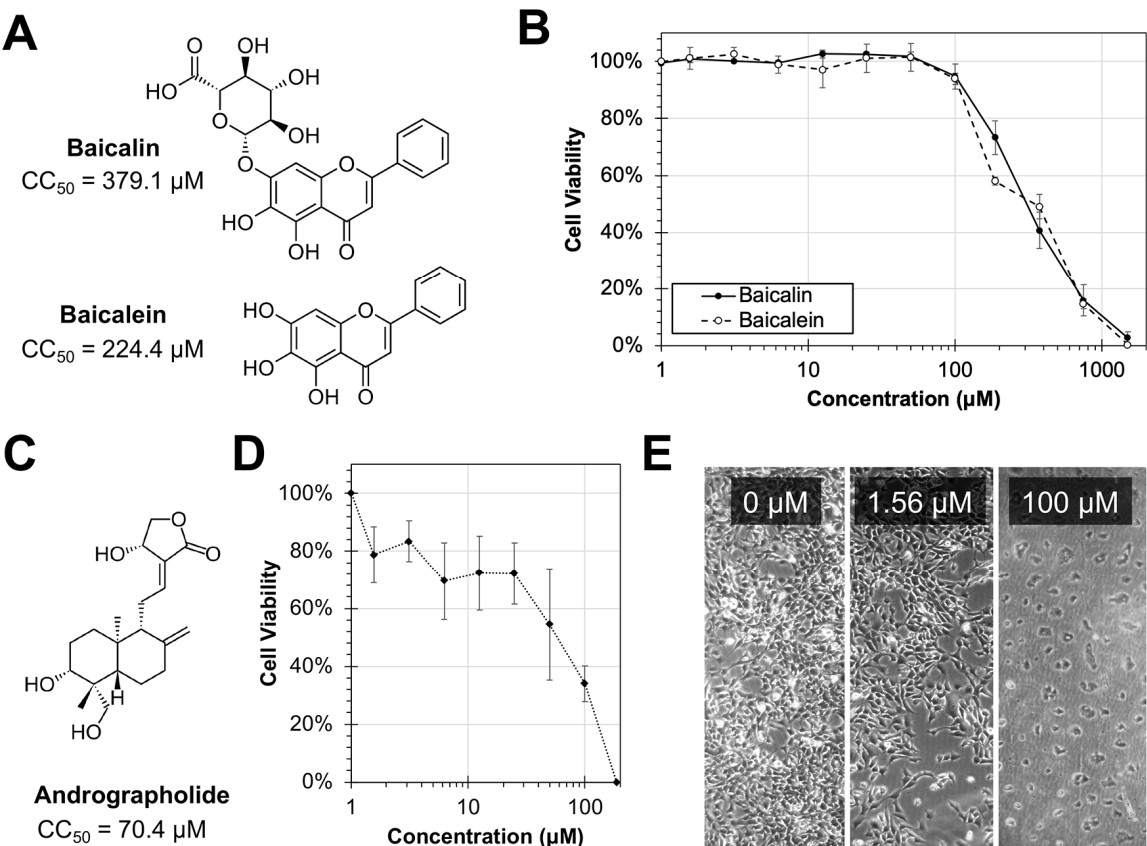

**Figure 4.** Evaluation of cytotoxicity of baicalin, baicalein, and andrographolide on DBT-9 cells. (**A**) The chemical structures of baicalein and baicalin and (**C**) andrographolide are shown with their corresponding cell cytotoxicity 50% (CC$_{50}$) values in DBT-9 cells. (**B**,**D**) DBT-9 cells were treated with differing doses of baicalein and baicalin (**B**) or andrographolide (**D**) for 24 h before cell viability was determined by trypan blue exclusion assays. The average cell viability (±s.e.m.) is shown for at least three experimental replicates. (**E**) Images were captured of andrographolide-treated DBT-9 cells after 24 h.

### 3.3. Antiviral Activity of Baicalin and Baicalein against H5-MHV-FFL and O5-MHV-FFL

Previously, baicalin and baicalein have been shown to specifically bind to and inhibit SARS-CoV-2 3CLpro protease [28]. To assess whether these compounds share inhibitory activity against the analogous 3CLpro proteases of human coronaviruses HKU1 and OC43, the H5-MHV-FFL and O5-MHV-FFL reporter strains were either pre-treated, co-treated, or post-treated with baicalin or baicalein (Figure 5).

The concurrent treatment of DBT-9 cells with both virus and inhibitor resulted in dose-dependent inhibition, albeit with distinct differences in susceptibility between the strains. The effective concentration that resulted in a 50% reduction in virus remaining (EC$_{50}$) was calculated for each combination of treatment and virus tested. H5-MHV-FFL exhibited a greater sensitivity to baicalein treatment with an EC$_{50}$ of 3.4 µM compared to an EC$_{50}$ of 48.0 µM during co-treatment with baicalin. In contrast, O5-MHV-FFL was more sensitive to baicalin (EC$_{50}$ = 8.0 µM) than baicalein (EC$_{50}$ = 68.9 µM).

The viruses, collectively, were less susceptible to either pre-treatment or post-treatment with baicalin or baicalein (Table 1). When comparing pre- to post-treated infections, the viruses were considerably more sensitive to pre-treatment than post-treatment. Most notably, there was very little inhibition (EC$_{50}$ > 100 µM) detected after post-treatment with either compound.

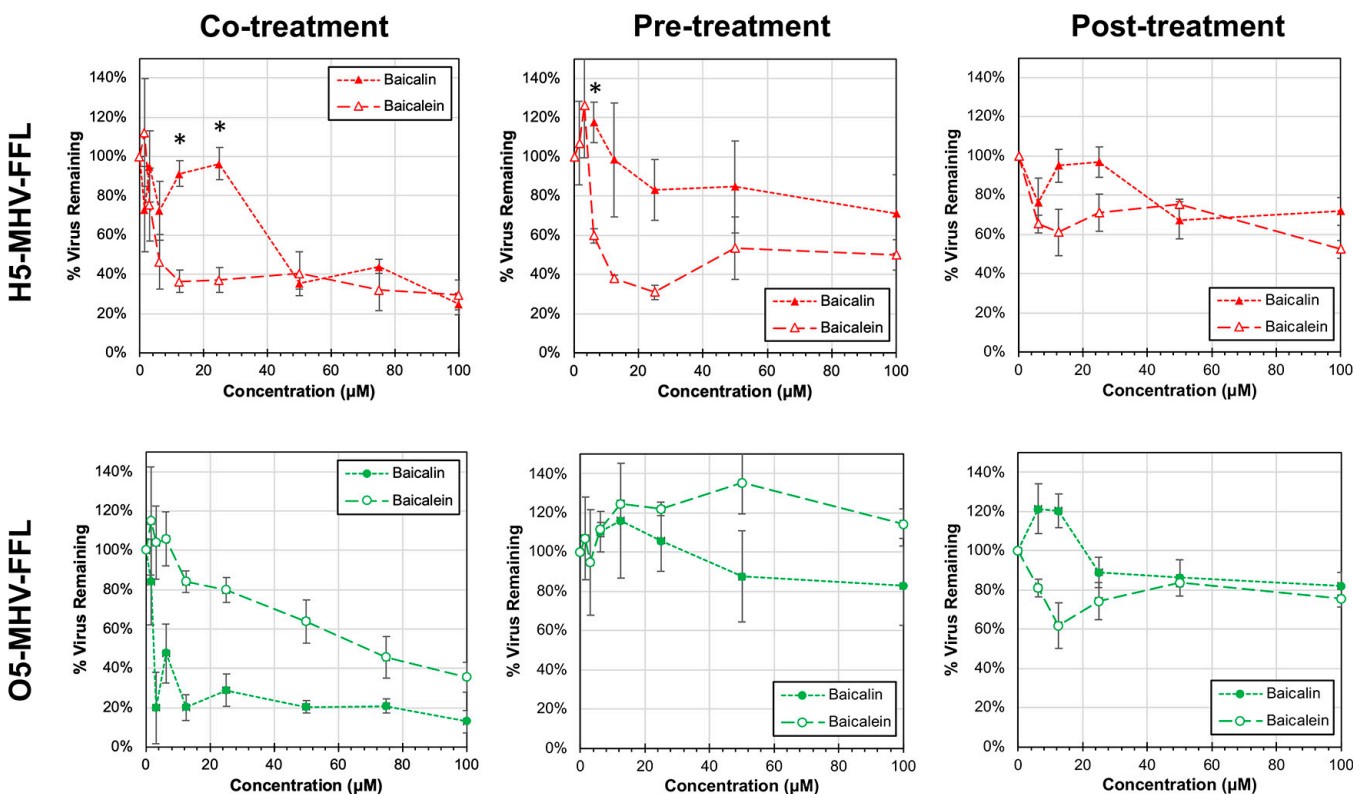

**Figure 5.** Antiviral activity of baicalin and baicalein against H5-MHV-FFL and O5-MHV-FFL. DBT-9 cells were either pre-treated (at 6 h prior to infection), co-treated, or post-treated (at 6 h post infection) with a range of concentrations of baicalin (filled symbol) or baicalein (unfilled symbol) during infections with an MOI of 0.01 of either H5-MHV-FFL (red) or O5-MHV-FFL (green). The data shown reflect the average percent of virus remaining ($\pm$s.e.m.) compared to no treatment of three experimental replicates. Statistical analyses were performed between baicalin and baicalein treatments for each concentration. Significant differences ($p < 0.005$; Student's *t*-test) are indicated (*).

**Table 1.** Cytotoxicity and antiviral activities of andrographolide, baicalin, and baicalein.

|  | Andrographolide [1] | Baicalin | Baicalein |
|---|---|---|---|
| Cytotoxicity ($CC_{50}$) | 70.4 μM | 379.1 μM | 224.4 μM |
| H5-MHV-FFL Inhibition | | | |
| Co-treatment $EC_{50}$ | - | 48.0 μM | 8.0 μM |
| Pre-treatment $EC_{50}$ | - | 183.3 μM | 11.5 μM |
| Post-treatment $EC_{50}$ | - | 203.3 μM | 111.2 μM |
| O5-MHV-FFL Inhibition | | | |
| Co-treatment $EC_{50}$ | - | 3.4 μM | 68.9 μM |
| Pre-treatment $EC_{50}$ | - | >300 μM | >300 μM |
| Post-treatment $EC_{50}$ | - | 224.0 μM | >300 μM |

[1] Was not tested for antiviral activity due to cytotoxic effects.

### 3.4. Identifying Potential Patterns of Viral Escape to Baicalein Treatment

In earlier studies, biochemical and structural analysis were performed on purified SARS-CoV-2 proteases to determine how baicalein was binding to and inhibiting the protease [28]. These studies and subsequent studies looking at SARS-CoV revealed that baicalein bound to the opening of the active site coordinating with several residues in and around the peptide-binding site [29]. Due to the observed differences in susceptibility between H5-MHV-FFL and O5-MHV-FFL to inactivation with baicalein, we passaged both

viruses in the presence of three times their $EC_{50}$ to promote viral escape and evaluate whether the viruses share common patterns of escape against the compound. Genetic sequencing of the nsp5-coding region for each of the two viruses was performed after five passages. No coding mutations were detected in the O5-MHV-FFL stocks; however, in H5-MHV-FFL, there was a tyrosine to serine change at amino acid residue 37 (Y37S) due to a change in the codon from U<u>A</u>U to U<u>C</u>U. Modeling the Y37S mutation onto the crystal structure of HCoV-HKU1 3CLpro shows that the mutation is located in domain 1 behind the active site of the protease (Figure 6).

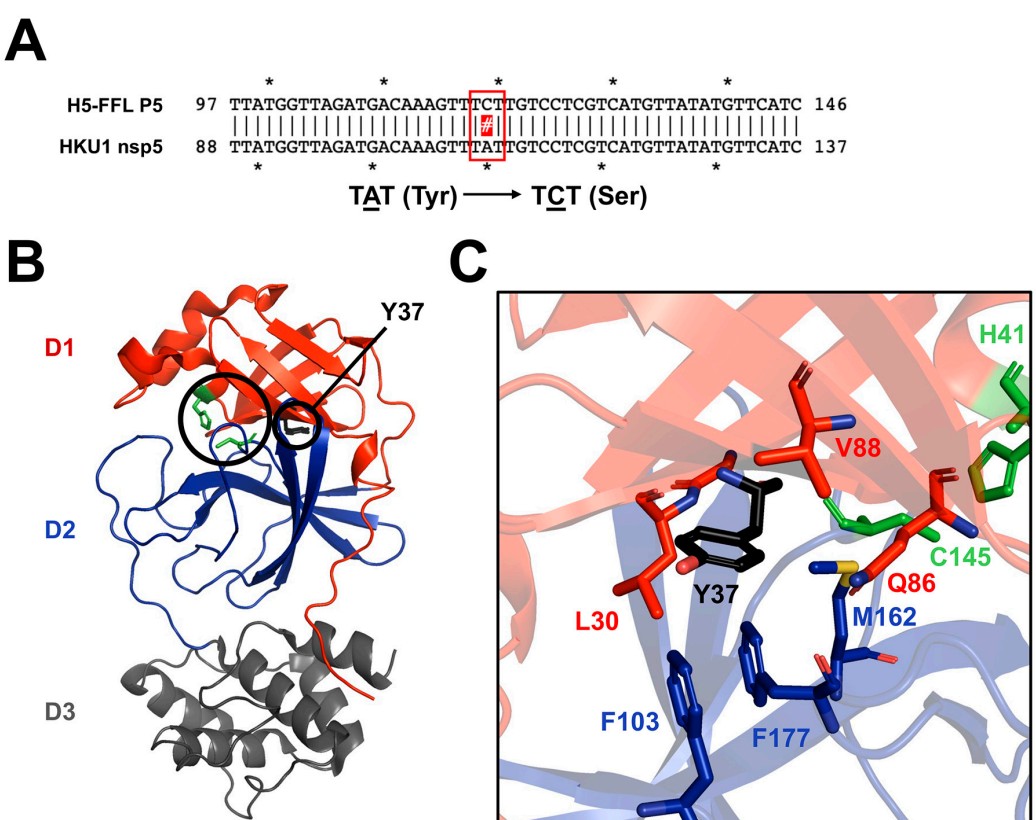

**Figure 6.** Y37S escape mutation detection in the HKU1 3CLpro protease of H5-MHV-FFL induced by baicalein. (**A**) Sequence alignment comparing the passage 5 (P5) mutant of H5-FFL and the wild-type HKU1 nsp5 gene sequence. The identified mutation is indicated by the box. (**B**) Structure of the 3CLpro protease of HCoV HKU1 (PDB 3d23). Domains 1 (D1; red), 2 (D2; blue), and 3 (D3; gray) are color-coded and labeled. The catalytic dyad residues H41 and C145 are shown in green, and a larger circle indicates their locations [18]. The Y37 residue location is also circled, and both are shown in black and labeled. (**C**) A view from behind the active site showing the pocket of side-chain residues surrounding the original Y37 amino acid. All amino acids shown (except catalytic residues shown in green) are within 4 Å of the mutated tyrosine residue. Images were obtained using PyMol 2.4.

## 4. Discussion

In this study, we describe the successful development of a pair of chimeric murine coronaviruses that express the 3CLpro proteases of human common-cold coronavirus strains HKU1 or OC43 as well as a firefly luciferase reporter for rapid inhibitor screening. Using these reporter viruses, we observed sensitivity to two plant-derived inhibitors of SARS-CoV-2 3CLpro, baicalin and baicalein. Previous studies have shown that baicalein binds to the face of the active site of SARS-CoV-2 protease [28]. Despite HCoV HKU1 and OC43 sharing only 48% and 47% of their amino acid sequence identities, respectively, with SARS-CoV-2, the crystal structures of HKU1 and SARS-CoV-2 3CLpro remain highly conserved, especially in domains 1 and 2, which form the active sites of the proteases (Figure 7).

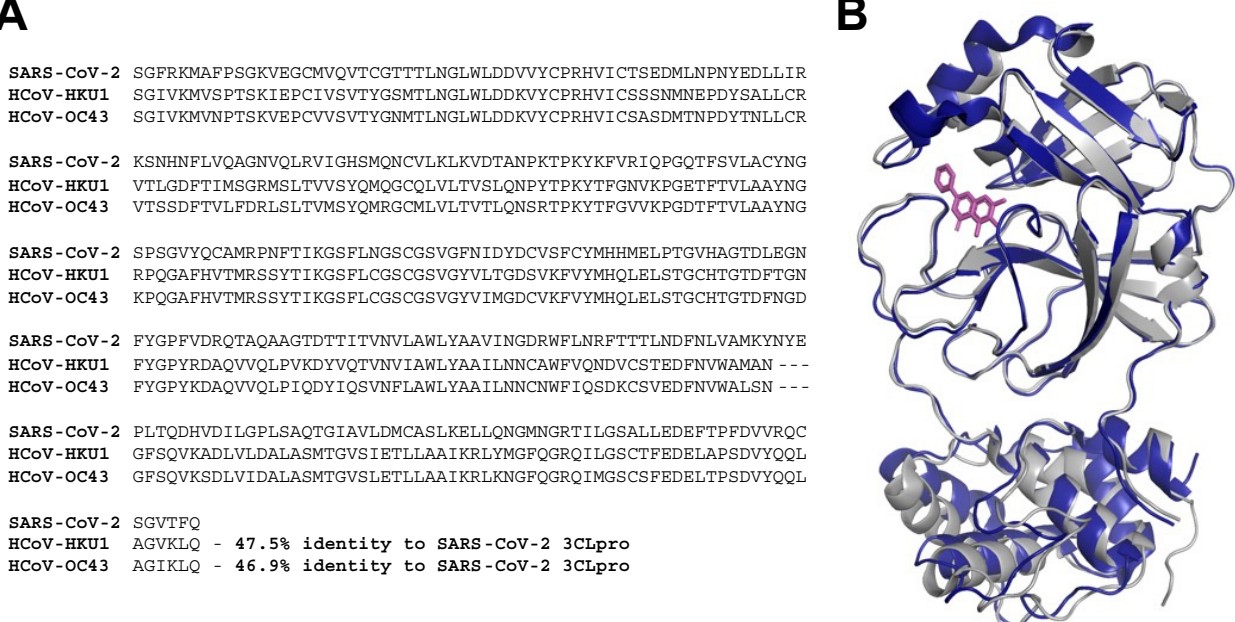

**A**

```
SARS-CoV-2  SGFRKMAFPSGKVEGCMVQVTCGTTTLNGLWLDDVVYCPRHVICTSEDMLNPNYEDLLIR
HCoV-HKU1   SGIVKMVSPTSKIEPCIVSVTYGSMTLNGLWLDDKVYCPRHVICSSSNMNEPDYSALLCR
HCoV-OC43   SGIVKMVNPTSKVEPCVVSVTYGNMTLNGLWLDDKVYCPRHVICSASDMTNPDYTNLLCR

SARS-CoV-2  KSNHNFLVQAGNVQLRVIGHSMQNCVLKLKVDTANPKTPKYKFVRIQPGQTFSVLACYNG
HCoV-HKU1   VTLGDFTIMSGRMSLTVVSYQMQGCQLVLTVSLQNPYTPKYTFGNVKPGETFTVLAAYNG
HCoV-OC43   VTSSDFTVLFDRLSLTVMSYQMRGCMLVLTVTLQNSRTPKYTFGVVKPGDTFTVLAAYNG

SARS-CoV-2  SPSGVYQCAMRPNFTIKGSFLNGSCGSVGFNIDYDCVSFCYMHHMELPTGVHAGTDLEGN
HCoV-HKU1   RPQGAFHVTMRSSYTIKGSFLCGSCGSVGYVLTGDSVKFVYMHQLELSTGCHTGTDFTGN
HCoV-OC43   KPQGAFHVTMRSSYTIKGSFLCGSCGSVGYVIMGDCVKFVYMHQLELSTGCHTGTDFNGD

SARS-CoV-2  FYGPFVDRQTAQAAGTDTTITVNVLAWLYAAVINGDRWFLNRFTTTLNDFNLVAMKYNYE
HCoV-HKU1   FYGPYRDAQVVQLPVKDYVQTVNVIAWLYAAILNNCAWFVQNDVCSTEDFNVWAMAN ---
HCoV-OC43   FYGPYKDAQVVQLPIQDYIQSVNFLAWLYAAILNNCNWFIQSDKCSVEDFNVWALSN ---

SARS-CoV-2  PLTQDHVDILGPLSAQTGIAVLDMCASLKELLQNGMNGRTILGSALLEDEFTPFDVVRQC
HCoV-HKU1   GFSQVKADLVLDALASMTGVSIETLLAAIKRLYMGFQGRQILGSCTFEDELAPSDVYQQL
HCoV-OC43   GFSQVKSDLVIDALASMTGVSLETLLAAIKRLKNGFQGRQIMGSCSFEDELTPSDVYQQL

SARS-CoV-2  SGVTFQ
HCoV-HKU1   AGVKLQ - 47.5% identity to SARS-CoV-2 3CLpro
HCoV-OC43   AGIKLQ - 46.9% identity to SARS-CoV-2 3CLpro
```

**B**

**Figure 7.** 3CLpro sequence alignment and structure overlay. (**A**) A primary sequence alignment generated using ClustalW of the 3CLpro proteases of human coronaviruses SARS-CoV-2, HKU1, and OC43. The percent sequence identity relative to SARS-CoV-2 is shown. (**B**) Overlay showing the crystal structures of SARS-CoV-2 (PDB 6M2N) in a complex with baicalein (shown in pink) and the structure of HCoV-HKU1 (PDB 3D23) [18,28].

Our data support that both baicalin and baicalein are targeting the 3CLpro of both human common-cold coronaviruses HKU1 and OC43. Previous studies using SARS-CoV-2 3CLpro have shown that baicalein and baicalin bind to the active site pocket of the protease [28,29]. Specifically, we observed strain-specific differences between our reporter viruses that only differ in their 3CLpro expression in the two compounds. Furthermore, when passaged under a high selection of baicalein, we identified an escape mutation behind the active site of HKU1 3CLpro. While the therapeutic potential for baicalin and baicalein remains limited in their current forms, the micromolar $EC_{50}$ values were found to be much lower than the corresponding cytotoxicities in DBT-9 cells and may warrant further modification and study. A distinct limitation to our study is that we were unable to perform a direct assay to demonstrate that baicalin and baicalein bind to and block the active sites of HKU1 and OC43; however, our collective data are consistent with this outcome. Moreover, our findings demonstrate the utility of these reporter viruses for the rapid screening of potential inhibitors to human coronaviruses in the safe and established MHV virus system.

With the recent emergence of SARS-CoV-2 triggering a worldwide pandemic, there is considerable concern for future emerging coronavirus infections. These studies support the continued development of chimeric platforms for the screening of coronavirus inhibitors against other common targets, including viral polymerase and its related replication machinery. The increased identification of novel animal and zoonotic coronaviruses argues for testing broadly applicable antivirals against both known and new zoonotic viruses [13,30,31]. The use of chimeric reporter viruses can overcome the challenges in safely cultivating these viruses. Additionally, antiviral compounds can be screened against the FFL reporter viruses in less than 24 h by assessing luciferase production, and any differences in sensitivity to the compounds may provide insight into divergent determinants of enzyme function across coronaviruses.

The sequence identity among genogroups is between 40 and 55% compared to MHV 3CLpro [16]. We have previously demonstrated the inability to recover chimeric viruses in the MHV background outside of the genogroup 2a of beta-CoVs. However, most

viruses share, within genogroups, a considerably higher sequence identity (often in excess of 80% for 3CLpro). Based on sequence identity, we would hypothesize that 3CLpro proteases could be exchanged among viruses in the same genogroup. We propose that "platform viruses" could be established for each genogroup for the chimeric exchange of 3CLpro or, potentially, other antiviral targets such as the papain-like proteases, the RdRp (nsp12), or novel enzymes such as nsp16-2′O-methyltransferase or nsp14-exonuclease-N7-methyltransferase. This would also have the advantage of potentially establishing virus backgrounds that do not require BSL3 or select agent management. However, previous attempts to recover MHV chimeras expressing more divergent 3CLpro proteases such as those from either SARS-CoV were unsuccessful [16]. While these findings would suggest that an MHV chimera expressing the recent emergent SARS-CoV-2 is unlikely, given its genetic similarity to SARS-CoV, there may be a possibility of generating a chimera based on a more similar zoonotic coronavirus background for testing. Thus, it might be possible to define several viruses in common cell types that could be used to represent the majority of known human and animal CoVs for testing in cell culture as well as rapid testing new emerging CoVs, particularly those with challenges in culture or restrictions in lab use. While this would not be a trivial undertaking, the continued SARS-CoV-2 epidemic demonstrates the importance of approaches to test for new inhibitors of CoVs.

**Author Contributions:** Conceptualization, E.R.H., Y.S.B., S.T., C.C.S. and D.C.B.; methodology, S.T., C.C.S. and D.C.B.; validation, E.R.H., Y.S.B., C.C.S. and D.C.B.; formal analysis, E.R.H., J.X.F., J.M.D., A.R.A., Y.S.B., C.M.H., J.D.B., M.N.T., C.C.S. and D.C.B.; data curation, E.R.H., J.X.F., J.M.D., A.R.A., Y.S.B., C.M.H., J.D.B., M.N.T., C.C.S. and D.C.B.; writing—original draft preparation, E.R.H., J.X.F., J.M.D., C.M.H., J.D.B. and S.T.; writing—review and editing, C.C.S. and D.C.B.; supervision, C.C.S. and D.C.B.; funding acquisition, D.C.B. and C.C.S. All authors have read and agreed to the published version of the manuscript.

**Funding:** Research support for this study came from a Butler University Holcomb Awards Committee research grant to C.C.S., summer research support via the Butler Summer Institute (BSI) for E.R.H., Y.S.B. and J.M.D., and funding provided by both the Butler University Department of Biological Sciences (C.C.S.) and the DeSales University Department of Biology (D.C.B.).

**Institutional Review Board Statement:** Not applicable.

**Informed Consent Statement:** Not applicable.

**Data Availability Statement:** Research data will be provided upon reasonable request.

**Acknowledgments:** We thank Mark Denison (Vanderbilt) for providing cells and virus stocks used in the experiments in this study. The Butler Student Virology Group includes the following students enrolled in the BI 440 Molecular Virology course (Spring 2022) that played a key role in performing pilot studies related to this work: Yara Batista, Letitia Bortey, Sara Breniser, Jon Brooks, Megan Fleshman, Christina Fukada, Keeley Hagan, Abby Heilman, Elise Huffman, Madison Kolanowski, Katie McCullough, Jordan Miles, Adia Mimms, Tara Poindexter, Aubrianna Radee, Megan Reed, and Jade Woo. We would like to thank other members of the Stobart and Beachboard labs and faculty members of their respective departments for their feedback and suggestions regarding both data acquisition and manuscript preparation.

**Conflicts of Interest:** The authors declare no conflicts of interest.

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
