# Peer review of "Development of Mouse Hepatitis Virus Chimeric Reporter Viruses Expressing the 3CLpro Proteases of Human Coronaviruses HKU1 and OC43 Reveals Susceptibility to Inactivation by Natural Inhibitors Baicalin and Baicalein"

_covid, doi:10.3390/covid4020016_

Round 1
Reviewer 1 Report
Comments and Suggestions for Authors
The authors develop and validate a reporter virus system containing luciferase-labelled 3CL proteases from two different human coronaviruses: HKU1 and OC43. Growth kinetics are assessed then tested against compounds previously reported to target the 3CL protease of SARS-CoV-2. Two of the compounds tested, baicalin and baicalein, exhibited micromolar efficacy against both chimeric reporter viruses. Finally, the authors conducted resistance studies using baicalin and baicalein and found one “escape mutant” in the OC43 chimera treated with baicalein. Though the authors present a useful tool for identifying broad-spectrum coronavirus protease inhibitors, support for their system and inhibitors are lacking in this manuscript. While the authors show kinetics and inhibition of chimeric viruses, the data presented lack statistical significance, sufficient number of biological replicates, and direct data demonstrating activity of the proteases studied to support their conclusions. Additional experiments and repeats are necessary.
Major comments:
-Few abbreviations throughout the text are defined
-Methods section requires significant revision (sequencing primers should be included, info on reagents, cell lines used etc need to be included).
-Statistical significance is lacking throughout, the authors need to address how stats were calculated and include them in both figures and results.
-Experiments throughout the manuscript lack an appropriate number of experimental replicates, additional repeats are needed for figures 3-5.
-While the chimeric viruses reported here will be a useful tool for the field, the authors need to show evidence the compounds tested are influencing 3CL protease activity directly as they claim in the discussion, not just inhibiting viral replication.
-Figure 3B. the authors state in the text in lines 196-207 that the growth kinetics between the chimeric viruses and WT are identical. No values or statistics are given to support this claim and it appears in figures 3A and 3B the H5-MHV-FFL virus has different growth kinetics from WT (in fig 3A the H5-MHV-FFL appears to grow more slowly than the other viruses and in 3B the final time point of H5 is one log higher than the WT FFL and O5 FFL reporters). Without statistical significance, the graph as is is not sufficient to support the authors claim that growth kinetics are identical.
-Figure 3B. WT-MHV only has one point on the graph at 16hpi. Why is this the only point for this virus indicated on the graph? And why is it registering 102 RLU if it does not contain a luciferase reporter (assuming the authors adjusted this graph for background)?
-The authors need to include sequence data for their escape mutant in figure 6.
Minor comments:
-It is unclear why andrographolide was excluded from antiviral efficacy studies outlined in table 1. Though a CC50 of ~70uM is not ideal, if the EC50 of the compound is low enough against the chimeras this could be another promising broad-spectrum compound.
-Since this study is based off of research done in SARS-CoV-2, the authors should consider using a reporter virus containing the SARS-CoV-2 protease as a control in the antiviral activity assays.
Comments on the Quality of English Language
Minor grammatical errors in the manuscript, quality is fine overall.
Author Response
We appreciate the thoughtful and detailed suggestions and recommended revisions by the reviewer. We have addressed all of these comments below along with the line numbers and/or figures that have been revised. Thank you for time and consideration.
Major comments:
-Few abbreviations throughout the text are defined
We have updated the text to define or clarify the following abbreviations: CoVs (line 32), 3CLpro (line 67), Mpro (line 67), and RLUs (line 214).
-Methods section requires significant revision (sequencing primers should be included, info on reagents, cell lines used etc need to be included).
We have updated the methods section to include the sources for reagents (lines 114 – 119; lines 145 – 151). The details for the non-commercially available DBT-9 and BHK-R cell lines used are included in the methods (lines 114 – 121). In addition, we have included the sequences for the primers used (lines 145 – 151).
-Statistical significance is lacking throughout, the authors need to address how stats were calculated and include them in both figures and results.
We have included statistical analyses where relevant for all figures. This includes the replication assays in Figure 3, differences between baicalin and baicalein for cytotoxicity in Figure 4, and between treatments for all conditions in Figure 5. In addition, we incorporated statements in the written results and figure legends referring to these statistical outcomes (lines 208 – 212; lines 215 – 217; lines 226 – 233; lines 247 – 248; lines 293 – 294).
-Experiments throughout the manuscript lack an appropriate number of experimental replicates, additional repeats are needed for figures 3-5.
Fig. 3B and 4D only had 2 replicates each. We have added an additional replicate, so all experiments throughout the manuscript now have a minimum of 3 replicates performed. As a result of these outcomes, we updated the CC50 for andrographolide based on the third replicate.
-While the chimeric viruses reported here will be a useful tool for the field, the authors need to show evidence the compounds tested are influencing 3CL protease activity directly as they claim in the discussion, not just inhibiting viral replication.
We hoped to test the activity on purified proteins where we could look at cleavage of a tag, however, we were unable to resolve the size shift when the tag is removed based on the assay we had available. We have plans to develop a cleavage assay using a FRET probe but didn’t not have time to complete this within the editor’s timeline. As such, we revised our discussion section to specifically address this limitation to our study (lines 357 – 360).
-Figure 3B. the authors state in the text in lines 196-207 that the growth kinetics between the chimeric viruses and WT are identical. No values or statistics are given to support this claim and it appears in figures 3A and 3B the H5-MHV-FFL virus has different growth kinetics from WT (in fig 3A the H5-MHV-FFL appears to grow more slowly than the other viruses and in 3B the final time point of H5 is one log higher than the WT FFL and O5 FFL reporters). Without statistical significance, the graph as is is not sufficient to support the authors claim that growth kinetics are identical.
We have performed statistical analyses to make pairwise comparisons at all time points between viruses and have revised our results section to more clearly address significant differences observed. Most notably, we did find that H5-MHV-FFL significantly differed from the non-FFL H5-MHV at several time points and removed mention of these viruses being identical by plaque assay (lines 208 – 212). We found no statistical differences based on luciferase activity (lines 215 – 217).
-Figure 3B. WT-MHV only has one point on the graph at 16hpi. Why is this the only point for this virus indicated on the graph? And why is it registering 102 RLU if it does not contain a luciferase reporter (assuming the authors adjusted this graph for background)?
We clarified the role of the WT-MHV reading in the written results (lines 218 – 220). This data point reflects the background RLUs prior to background subtraction.
-The authors need to include sequence data for their escape mutant in figure 6.
We have now included the codon change rather than just the amino acid substitution in the written results (lines 315 – 317).
Minor comments:
-It is unclear why andrographolide was excluded from antiviral efficacy studies outlined in table 1. Though a CC50 of ~70uM is not ideal, if the EC50 of the compound is low enough against the chimeras this could be another promising broad-spectrum compound.
Our primary concern with performing viral assays with andrographolide stemmed from the apparent changes in cell morphology at all concentrations. We were concerned that any impact on virus replication could be confounded by these cellular changes rather the inhibitor itself. We clarified this position in the results and have included larger images of the cell morphology in Fig 4E for readers (lines 249 – 261).
-Since this study is based off of research done in SARS-CoV-2, the authors should consider using a reporter virus containing the SARS-CoV-2 protease as a control in the antiviral activity assays.
We recognize this would be a valuable control. However, as demonstrated in Stobart et al. 2013, we have previously tried to recover a chimera of MHV expressing 3CLpro of SARS-CoV (Betacoronavirus group 2b) and were unsuccessful. Additionally, CoV-HKU4 (Betacoronavirus group 2c), and the alphacoronaviruses HCoV-NL63 and HCoV-229E were also not able to be recovered in the background of MHV. Only HKU-1 and OC43 (both in betacoroanvirus group 2a with MHV) chimeras could be recovered. As such, we would not expect it to be possible to recover a virus containing the SARS-CoV-2 (Betacoronavirus group 2b) protease. We have included this information in the discussion and propose the use of alternative zoonotic viruses of a more close genetic background for future chimeric design (lines 383 – 393).
Reviewer 2 Report
Comments and Suggestions for Authors
The authors present a well written paper that is rather methodical than scientific. The paper is logical. The authors have created, tested and characterized a chimeric reporter system based on mouse hepatitis virus expressing the firefly luciferase and the 3CLpro proteases of human coronaviruses HKU1 and OC43. They have also tested three natural inhibitors originated from plants - baicalin, baicalein and andrographolide - and revealed susceptibility of human coronaviruses HKU1 and OC43 to both baicalin and baicalein. The andrographolide inhibitor occurred to be highly cytotoxic to cells, and thus was not further tested for antiviral activity.
As the authors say in the Introduction and Discussion sections there are plenty of coronaviruses circulated in the human population nowadays. And we do not know which virus might start a new pandemics. The reporter system they describe may be useful to test for new inhibitors of coronaviruses, and importantly, do not require BSL3.
Author Response
We would like the thank this reviewer for taking the time to review our article and provide us feedback.
Reviewer 3 Report
Comments and Suggestions for Authors
Revision manuscript “Development of mouse hepatitis virus (MHV) chimeric reporter viruses expressing the 3CLpro proteases of human coronaviruses HKU1 and OC43 reveal susceptibility to inactivation by natural inhibitors baicalin and baicalein”
In this manuscript, the authors tested the inhibitory activity of three natural compounds baicalin, baicalein and andrographolide against the 3CLpro proteases of CoV-HKU1 and CoV-OC43. All three compounds showed inhibitory activity against 3CLpro of both human coronaviruses, but andrographolide demonstrated cytotoxicity and failed to demonstrate selective toxicity towards the viruses.
The experimental design is well presented, and the conclusions are supported by the results. This experimental approach might be useful for assessing preliminarily the therapeutic potential of new antiviral molecules.
Author Response

(The authors gave the same response as above.)

Round 2
Reviewer 1 Report
In this manuscript, Huffman et al develop chimeric coronavirus 3CL protease reporters, validate growth kinetics and assess them against two protease inhibitors originally identified against SARS-CoV-2. The revised manuscript has significantly improved, and data and conclusions are presented clearly.
Lines 296-298 Baicalein resistance mutant. The addition of the codon information is appreciated, however could the authors please provide nsp5 sequence data comparing WT and mutant virus outside of the single codon indicated in the text (i.e. an alignment of a few dozen bases of sequencing data) to demonstrate Y37S is the only mutation that occurred in the HKU1 3CL protease region sequenced during the resistance study. Indicating a point mutation alone without additional sequence data surrounding it does not provide enough support for the conclusion made by the authors.
Author Response
We have amended Figure 6 (and its corresponding legend) to include the region of the sequence alignment containing the point mutation of the passaged mutant compared to the wild-type HKU1 nsp5 mutation. We thank the reviewer for this specific request as we discovered that the wrong codon change (still resulting in the reported Y37S mutation) was described in the text of the results. This was corrected as well. There were no other mutations observed in the sequencing data throughout the rest of the nsp5 coding region.